# Mechanisms of Neuroinvasion and Neuropathogenesis by Pathologic Flaviviruses

**DOI:** 10.3390/v15020261

**Published:** 2023-01-17

**Authors:** Liset de Vries, Alfred T. Harding

**Affiliations:** 1School of Life Sciences, Utrecht University, 3584 CS Utrecht, The Netherlands; 2Institute for Medical Engineering and Sciences, Massachusetts Institute of Technology, Cambridge, MA 02139, USA

**Keywords:** flavivirus, neuroinvasion, neuropathogenesis

## Abstract

Flaviviruses are present on every continent and cause significant morbidity and mortality. In many instances, severe cases of infection with flaviviruses involve the invasion of and damage to the central nervous system (CNS). Currently, there are several mechanisms by which it has been hypothesized flaviviruses reach the brain, including the disruption of the blood–brain barrier (BBB) which acts as a first line of defense by blocking the entry of many pathogens into the brain, passing through the BBB without disruption, as well as travelling into the CNS through axonal transport from peripheral nerves. After flaviviruses have entered the CNS, they cause different neurological symptoms, leading to years of neurological sequelae or even death. Similar to neuroinvasion, there are several identified mechanisms of neuropathology, including direct cell lysis, blockage of the cell cycle, indication of apoptosis, as well as immune induced pathologies. In this review, we aim to summarize the current knowledge in the field of mechanisms of both neuroinvasion and neuropathogenesis during infection with a variety of flaviviruses and examine the potential contributions and timing of each discussed pathway.

## 1. Introduction

### 1.1. Background

Viruses of the flavivirus genus are present on every continent. Examples of flaviviruses that can cause severe symptoms in humans are mosquito-borne viruses, such as dengue virus (DENV), Zika virus (ZIKV), West Nile virus (WNV), yellow fever Virus (YFV), and Japanese encephalitis virus (JEV), as well as several tick-borne flaviviruses, including tick-borne encephalitis viruses (TBEV) and Powassan Virus (POWV) [1]. Although the clinical presentation differs among the described viruses, they all cause significant morbidity and mortality. DENV, for instance, can result in capillary leakage and hemorrhage that could prompt circulatory shock [1,2]. Similarly, YFV can cause hemorrhage and shock but also severe liver damage resulting in jaundice [1,3]. In addition, many severe cases of infection with flaviviruses are often associated with neurological disease [4,5].

Although less common than the systemic symptoms, DENV can have substantial neurological involvement, such as encephalopathy, Guillain–Barré syndrome, and acute disseminated encephalomyelitis [6,7]. Other flaviviruses, such as ZIKV, TBEV, WNV, and JEV, however, are well-known for their involvement with the central nervous system (CNS). JEV and WNV cause parkinsonian symptoms with sequelae that can last years after the initial infection [8], whereas TBEV mainly causes meningitis and meningoencephalitis [9]. For decades, ZIKV was only reported to cause a mild febrile illness in infected patients. However, new outbreaks occurred in 2007 on the Yap islands and subsequently spread to South America [10]. Since then, it has been shown that ZIKV has the ability to cross the placental barrier in pregnant women, causing alarming new neurological symptoms in the developing fetus such as microcephaly and other fetal abnormalities, commonly called congenital Zika syndrome [11,12]. Together, flavivirus infections cause significant morbidity and mortality annually, emphasizing the urgent need for effective vaccines and treatments.

The best way to prevent these severe infections is to vaccinate at-risk populations. Vaccines are available against JEV, TBEV, and YFV, dramatically decreasing the mortality of these infections. Currently, vaccines against WNV, DENV, and ZIKV are in clinical trials or conditionally approved [13]. Despite these advances, flavivirus infections are still widespread. DENV for instance, infects more than 96 million people annually [2]. The search for an effective vaccine for DENV is complicated by the possibility of antibody-dependent enhancement (ADE), which calls for a tetravalent vaccine that produces neutralizing antibodies against all four serotypes simultaneously [14]. Furthermore, several relatively unknown flaviviruses such as Usutu virus (USV) and Murray Valley Encephalitis Virus (MVEV) could emerge to cause a new global health threat [1]. The risk of zoonotic emergence of new flaviviruses and the continued threat of current endemic flaviviruses, despite available vaccines, shows the need for further research about flavivirus biology. In this review, we aim to summarize current findings on the mechanisms of how flaviviruses cross the blood–brain barrier to establish infection, as well as causes of cellular pathology once they have reached the brain.

### 1.2. Viral Migration during Infection and the Blood–Brain Barrier

Infection with flaviviruses occurs largely through mosquito or tick bites. As such, the first cells that are infected are located in the dermis, such as dermal macrophages and tissue resident dendritic cells (DCs), called Langerhans cells [15]. After local proliferation in the dermis and DCs, the virus can disseminate in different ways: by being released into the bloodstream as cell-free virus particles, or by being transported to the lymph nodes by infected DCs. Through this dissemination, the flavivirus can reach and infect secondary organs such as the lymph nodes, spleen, kidney, or liver [16]. Additionally, the brain can be infected, which can happen immediately after the first dissemination of virus, or later in the infection timecourse after multiplication of the virus in secondary organs.

After the initial infection, flaviviruses can invade the brain. The human body contains several defenses aimed at preventing viruses and other microorganisms from infecting the brain, chief of which is the blood–brain barrier (BBB). The BBB is a barrier between blood in the blood vessels and the neuronal, glial, and other cells residing in the brain, regulating homeostasis and protecting the brain from pathogens. Blood vessels have differing morphologies throughout the body. In general, they have three layers: the tunica intima, containing the endothelial cells laying on a basement membrane (BM), the tunica media with smooth muscle cells, and the fibroelastic tunica adventitia [17]. Capillaries do not have a tunica media or a tunica adventitia. They consist of a single layer of endothelial cells, a BM, and pericytes [17]. Importantly, capillaries of the brain are not fenestrated, meaning they possess an uninterrupted BM [18].

The non-fenestrated endothelial cell is not the only structural characteristic in the BBB providing protection. Blood in the vessels is sealed off from the vascular network in between endothelial cells by several attachment molecules that form tight junctions (TJ), preventing pericellular flow of fluid and molecules. Endothelial cells are maintained by pericytes and the vascular integrity is supported by astrocytes [19]. Additionally, the BM is complete and enfolds the capillary, making a division between the endothelial cells and the pericytes, as well as between the pericytes and the astrocytes. This is especially pronounced in the post-capillary venule, containing a perivascular space to regulate leukocyte trafficking. To ensure that no infected cells or pathogens cross the barrier, there are very low levels of leukocyte adhesion molecules and transcytosis [19]. These protective measures of the BBB at the same time limit the influx of nutrients necessary for the normal function of the brain. To overcome this, the endothelial cells contain transporter molecules.

There are several ways for neurotropic viruses to enter the CNS. The first and most commonly hypothesized way is by going through the BBB. This means that the virus needs to cross the BBB, either by breaking the BBB down or by passing through the endothelial cells. In the case of flaviviruses, this is considered plausible because flavivirus infection is oftentimes systemic, meaning that the virus has access to the bloodstream and can thereby be transported through blood vessels into the CNS [20]. The second major possibility for CNS entry is by axonal transport, which has been reported extensively for several other neurotropic viruses [21,22]. In this case, a neuron that spans a long distance from the periphery to the CNS would become infected in the periphery. The axon would provide a mode of transport directly to the CNS, without having to overcome the BBB obstacles. In this review, the possible mechanisms used by flaviviruses to gain access to the CNS are discussed, based on the most important human pathogenic flaviviruses.

After the flavivirus has entered the CNS by one of the two previously described methods, it can cause neuropathogenesis. The exact cause of the different symptoms triggered by different neuropathogenic flaviviruses remains debated. Here, we describe three ways in which flavivirus neuroinvasion can lead to neuropathogenesis. This includes not only neuronal death after direct infection but also the effects of the immune system combating the virus and the consequences of viral invasion for non-infected bystander cells.

## 2. Blood–Brain-Barrier Crossing

### 2.1. Mechanisms of Permeabilizing the BBB

As mentioned before, the BBB has an important function in protecting the CNS from pathogens. However, neurotropic viruses have developed a variety of ways in which to overcome this natural host barrier. One of the most well-known mechanisms by which neurotropic viruses overcome this obstacle is by directly targeting the BBB and damaging its integrity, shown in vivo in mice following JEV infection [23,24]. Several different pathways resulting in BBB breakdown are induced during flavivirus infection [25]. One of the most potent pathways responsible for damaging BBB integrity is the triggering of the production of matrix metalloproteases (MMP), which can cleave TJ proteins and collagen IV (Figure 1) [26]. It was shown in vivo that WNV infection leads to an upregulation of systemic MMP levels, which coincided with a reduction in TJ proteins as well as BBB disruption [27]. This process can be accomplished by causing oxidative stress, which can lead to the activation of MMPs. JEV infection was shown to similarly upregulate MMP-9 in an ROS-dependent manner in astrocytes [28]. Interestingly, astrocytes are not the only cell type involved in this process as DENV was reported to infect DCs, also resulting in an upregulation of MMP secretion [29]. Studies with WNV demonstrated that macrophages infected during the first and systemic phases of the infection timecourse also upregulate MMPs that can cleave TJ molecules and enhance BBB permeability [30,31]. The critical role of MMPs in BBB breakthrough was further supported by the observation that MMP knockout mice exhibited a lower amount of BBB permeability during WNV infection [31].

It is important to note, however, that other reports have demonstrated that macrophages can also provide a protective effect against neuroinvasion, although this can also be attributed to the protection macrophages provide against peripheral replication [32]. Similarly, CD11c^hi^ DC ablation was shown to aggravate JEV neuroinvasion and BBB permeability, possibly due to higher systemic inflammation after peripheral replication [33]. It is also worth noting that the breakdown or downregulation of TJ molecules by flaviviruses is not restricted to the tissues of the BBB. ZIKV, for example, was shown to utilize a similar process to induce proteasomal degradation of ZO-1 proteins in the placental barrier [34].

A second reported mechanism for BBB breakdown is the activation of pathways downstream of toll-like receptor 3 (TLR3). TLR3 is a pattern-recognition receptor that recognizes double-stranded RNA (dsRNAs) and induces an anti-viral response [35]. During flavivirus genomic replication dsRNAs, termed replication intermediates, are generated and can be detected by this receptor. In general, TLR3 activation has an anti-viral effect by triggering innate anti-viral immunity and the secretion of pro-inflammatory cytokines, thereby recruiting immune cells to combat the infection [36]. Recent data indeed showed a protective effect of TLR3 against neurological disease and BBB permeability by JEV [37]. This protective effect of TLR3 was also reported by Daffis and colleagues during WNV infection, although this effect was mostly seen after intracranial injection, suggesting TLR3 can also improve outcomes post neuroinvasion [38]. TLR3 activation, however, was also reported to induce BBB leakage and thereby enhance virus influx in the brain. This was confirmed by the disruption of Claudin-5 expression following Poly(I:C), a synthetic dsRNA compound, stimulation of the TLR3 receptor [39]. Contrary to the protective effect of TLR3 described by Daffis and colleagues, other reports showed that TLR3 knockout mice had a higher resistance to lethal WNV infection compared to WT mice [38,40]. In considering these opposing findings, it has been suggested that the role of TLR3 in the flavivirus immune response changes throughout the different stages of infection. The immune response TLR3 induces appears to provide a protective role for neurons and peripheral tissues during infection, but causes damage and increased permeability at the BBB [41]. Next to triggering the TLR3 response, direct infection of HBMECs with Dengue virus was also shown to induce ROS, resulting in HBMEC death and higher endothelial permeability. Additionally, non-mitochondrial ROS was shown to increase cytokine secretion by the infected HBMECs [42]. Importantly, while TLR3 stimulation and other innate pathways have been correlated with BBB permeability, the question remains as to whether this is a direct result of TLR3 signaling in these tissues or a secondary mechanism resulting from the produced inflammatory cytokine and chemokines.

Along this line of questioning, it has been shown that the release of pro-inflammatory cytokines and chemokines causes BBB disruption [43,44]. For example, IFNα, IL-6, and IL-8 were reported to affect BBB permeability [45]. Cytokine and chemokine levels in both serum and CSF fluid are upregulated during flavivirus infection, prompting the question if this provides the virus with an advantage for neuroinvasion through a permeable BBB [46]. During JEV and DENV infection, patients with higher inflammatory cytokine levels experienced worse neuropathological outcomes, substantiating this hypothesis [46]. This is exemplified in a report where JEV-infected pericytes were reported to produce IL-6, causing proteasomal degradation of BBB protein ZO-1 and subsequent leakage [47]. Importantly, however, some cytokines, such as IFNλ, have been reported to tighten the BBB, thereby restricting WNV neuroinvasion [48]. Additionally, cytokines play an essential role in the protection of neurons against viral infection and replication [49]. An example of the protective role of cytokines is IFNα, which is essential for individual neuron and astrocyte cell protection against the cytopathic effects of viral replication [50]. These reports of BBB tightening and neuron protection by cytokines, together with the opposing mechanism of BBB disruption, show that the role of cytokines in flavivirus pathogenesis likely represents another example of the balancing act of regulating inflammation during viral infection to restrict the virus without causing excessive host damage.

Although cytokines and chemokines were shown to be causative molecules for BBB breakdown in vitro, there could also be a secondary mechanism by which they cause BBB permeabilization. Namely, cytokines and chemokines recruit leukocytes, which can induce BBB breakdown themselves. Peripheral leukocytes have limited access to the CNS in a normal situation. However, during infection of the endothelial cells in the blood vessel lining, leukocyte adhesion proteins and chemokines will be upregulated, causing leukocytes to be recruited and migrate through the BBB [51]. ICAM-1, an adhesion protein responsible for the attraction and binding of lymphocytes, was reported to be significantly upregulated after WNV infection [52]. Interestingly, in this report the researchers found that ICAM-1 presence appeared to benefit WNV neuroinvasion as ICAM-1 knockout mice were more resistant to lethal WNV encephalitis [52]. Additionally, in the brains of WT mice WNV titer and BBB permeability were significantly higher than in the brains of ICAM-1 KO mice, further supporting a role for BBB breakdown in this process [52]. ICAM upregulation coinciding with higher BBB permeability was also observed in ZIKV-infected mice, although the BBB permeability was not as pronounced as with WNV infection [53]. This BBB disruption was most likely caused by the recruitment of leukocytes, which can cause permeability by migrating through the BBB [54]. An alternative explanation would be that these higher rates of neuroinvasion result from virus passing the BBB inside of infected leukocytes. This mechanism is called the “Trojan horse” and will be discussed at a later point. Putting this into perspective, was a report stating that neutrophil invasion of the CNS was not paired with BBB breakdown in TBEV patients [55]. Nevertheless, there is a large body of evidence pointing towards leukocyte invasion of the CNS being an additional factor in BBB breakdown.

Migration through the BBB is not the only way that leukocytes can contribute to the process of viral breakdown of the BBB. Flaviviruses can also trigger leukocytes to secrete BBB-disrupting molecules, such as proteases. Recently, JEV and DENV were found to induce mast cell secretion of the serine protease chymase, which cleaves BBB proteins, thereby breaking down the BBB [56,57]. Serum chymase levels were predictive of DENV disease severity, although this is possibly due to TJ disruption in tissues other than the brain [58].

After the BBB is permeabilized through one of the above-mentioned mechanisms, paracytosis can take place (Figure 1). Paracytosis describes the passing of virus or molecules through the space in between cells of the endothelial barrier [59]. The expression of TJ molecules in the BBB normally impedes paracellular infiltration. However, the previously described BBB breakdown mechanisms can break open the path for cell-free virus to pass the endothelial barrier. Indeed, DENV cell-free virus particles were described to pass through the BBB [60]. This mechanism does not only happen at the BBB, but also the placental barrier where it was reported that the breakdown of TJ proteins allowed the paracellular passage of ZIKV [61]. In conclusion, although BBB breakdown is not likely to encompass the only CNS invasion mechanism, evidence suggests that it heavily contributes to the wide-scale invasion of the CNS by flaviviruses.

### 2.2. Mechanisms of Passing through an Intact BBB

Although the conventional way viruses were thought to cross the BBB is through breakdown of the barrier itself, it has been found that flaviviruses can invade the CNS before BBB disruption or leakage occurs, suggesting that BBB breakdown is not the only mechanism by which flaviviruses invade the CNS [62,63]. For example, Tembusu virus, an avian encephalitic flavivirus, was recently detected in the brains of ducklings before any BBB disruption was observed [64]. The BBB breakdown happened simultaneously with the appearance of neurological clinical signs at a late stage of the infection, suggesting BBB breakdown was necessary for large-scale neuroinvasion, but it was not the first mechanism for neuroinvasion [64]. Indeed, it was found that inflammatory molecules such as IL-6, CCL5, IFNα, and CXCL10 were significantly upregulated during in vitro JEV infection of HBEC and astrocyte co-cultures, resulting in an increase in BBB permeability after neuroinvasion [65]. However, determining whether BBB breakdown is the first cause of neuroinvasion remains problematic, as it is difficult to gather BBB permeability data at different time points during in vivo studies. Moreover, a recent non-human primate (NHP) infection with ZIKV resulted in both acute and chronic BBB permeability, further highlighting the complexity of uncovering the timeline of neuroinvasion mechanisms [66].

The delay in BBB permeability in some studies suggests that disruption of the BBB is potentially responsible for large-scale invasion, but not the first entry of flaviviruses to the CNS. This would mean that there are other entry mechanisms. The first barrier that other entry mechanisms need to overcome to invade the CNS is the endothelial cell layer. An important way to overcome this barrier is by going through these cells. This can happen in two ways: transcytosis or infection of the endothelial cells themselves (Figure 2).

Transcytosis is the mechanism where molecules, or in this case viruses, are transported across an endothelial cell layer by intracellular vesicles. The virus is endocytosed by receptor-mediated or random sampling, through the formation of clathrin-coated or caveolae-mediated vesicles [67]. Instead of being sorted for degradation or recycling, the vesicle continues to the other side of the cell [67,68]. Endothelial cells in the vesicle walls of the CNS are polarized, ensuring that endocytosed vesicles go to one specific side of the cell [69]. After the vesicle with virus has crossed the cell, the virus is released at the other side by the fusion of the vesicle with the cell membrane.

Several studies indicate that transcytosis might be an important pathway by which flaviviruses enter the CNS. A first hint at this was shown with electron microscopy, where JEV was observed in coated and uncoated vesicles in endothelial cells and pericytes [70]. However, it could not be concluded from these pictures whether the virus was actively replicating in the cell or passing through by transcytosis. This distinction was later made in the work of Hasebe, R. et al., where they found that WNV-like particles could cross endothelial cells via this mechanism [71]. These particles were replication-deficient, indicating that transcytosis without active replication might be one mechanism by which BBB crossing occurs. In addition, ZIKV was shown to cross BBB cells in a transwell system without causing permeability [34]. Single-virus tracking techniques confirmed that this crossing happened through transcytosis, which was temperature-, microtubule-, and endocytosis-dependent [34]. It is important to note that a potential limitation of this research is that it is often performed using in vitro models, which do not represent the full BBB, and might therefore miss the complexity of an in vivo system [71].

Due to the limitations of in vitro research, in vivo research will always be needed to provide an accurate picture of the importance of flavivirus transcytosis through the BBB during infection. Jia Zhou et al. found an indication of transcytosis playing a role in flavivirus CNS involvement in a mouse model by showing downregulation of Mfsd2a by ZIKV [72]. Under normal circumstances, Mfsd2a suppresses transcytosis by regulating the amount of unsaturated fatty acids in the outer leaflet of the plasma membrane, which causes a blockade of caveolae-mediated transcytosis [67,73,74]. ZIKV E-protein was shown to promote ubiquitination and subsequent degradation of Mfsd2a [72]. It has been demonstrated that loss of Mfsd2a promotes transcytosis [73]. In addition to its role in transcytosis, Mfsd2a is also suggested to be important for ZIKV pathogenesis. When the effect of Mfsd2a downregulation was countered with the direct administration of Docosahexaenoic acid (DHA), an essential fatty acid that is normally taken up by Mfsd2a, the number of ZIKV copies in the brain was reduced and brain health increased [72]. It has to be noted that although ZIKV-induced brain injury was significantly lower after DHA administration, this could also be caused by the general benefits of DHA for brain development [72].

Moreover, Nakayama and colleagues recently showed that one of the differences between a virulent strain (MR766) and a less virulent strain (PRVABC59) of ZIKV was a difference in the ability to enter the CNS. This difference in mortality was eliminated by intracranial injection, showing the importance of crossing the BBB has for virulence [75]. From this work, it was proposed that the more virulent strain crosses the BBB with transcytosis, as there was more virus in the lower compartment in the in vitro setup, representing the intracranial side of the BBB, despite the endothelial cell monolayer remaining intact. Importantly, this finding does not, however, examine the role differential immune response might play, or exclude the possibility of viral replication in endothelial cells, as it was shown that ZIKV infection and replication in endothelial cells does not necessarily lead to cytopathic effects and subsequent loss of barrier function [76].

Transcytosis is partly responsible for BBB crossing and infiltration into the CNS. However, another mechanism to pass the BBB without damaging it is through active infection and replication in endothelial cells and subsequent viral release on the CNS compartmental side (Figure 2) [76,77]. This mechanism is possible without rupturing the BBB because flaviviruses are able to infect some cell types without causing cytopathic effects [76]. TBEV, ZIKV, Usutu virus, and POWV, for example, were shown in vitro to infect and replicate in human brain microvascular endothelial cells (hBMECs), albeit with different efficiencies and timing [76,77,78,79]. After Powassan infection, the absence of barrier disruption and the enhanced titer at the basolateral side, correlating with the intracranial side of the BBB, indicated a preference for virus release at that side [79]. Although there was no barrier disruption measured during the timepoints taken in this study, infection of hBMECs likely causes a release of inflammatory factors, leading to delayed, indirect barrier disruption via previously discussed mechanisms. Indeed, both WNV and ZIKV infection in hBMECs in vitro resulted in increased CCL5 chemokine production, which, as discussed before, can disrupt BBB integrity [76,80].

The most likely scenario of flavivirus crossing the BBB without obvious disruption would be a combination of both transcytosis as well as endothelial infection. This was shown by treating ZIKV-infected hBMECs with Nystatin, Chloroquine, and Brefeldin A (BFA) [63]. Nystatin can inhibit caveolae-mediated transcytosis, whereas BFA changes vesicle traffic from the ER to the Golgi, thereby inhibiting exocytosis after viral replication [81,82]. Chloroquine raises the pH of acidic compartments, taking out the trigger for E protein conformational changes, thereby inhibiting viral replication. All three compounds were able to substantially reduce virus crossing to the abluminal side of the cells, indicating that both transcytosis and endothelial cell infection play a key role in BBB crossing for ZIKV [63].

### 2.3. Trojan horse

The last option for BBB passage is a phenomenon called the “Trojan horse principle” (Figure 3). As previously discussed, immune cell migration forms an important mechanism for BBB disruption. However, it can also provide the virus with a different method of neuroinvasion, which happens when immune cells that are infected in the periphery enter the CNS. Recently it was shown that ZIKV, WNV, and JEV can employ the Trojan horse mechanism to enter the CNS by infecting monocytes that subsequently migrate through the BBB [83,84,85]. This was also shown in vivo where ZIKV infection in myeloid cells had a big impact on neuropathology [86]. Although several different factors are necessary for immune cell migration, there is one protein that was specifically implicated in both JEV as well as ZIKV pathogenesis: High-mobility group box 1 (HMGB1) [83,85]. HMGB1 is secreted by JEV-infected hBMECs and ZIKV-infected monocytes. HMGB1 can act as a chemoattractant and adhesion factor, and can thereby stimulate immune cell migration into the CNS [87]. Additionally, ICAM-1 and VCAM were induced by WNV infection in endothelial cells, providing a point of engagement for infected immune cells [54].

It is important to note, however, that the Trojan horse mechanism cannot occur on its own. In order for immune cells to migrate and cause this “Trojan horse” invasion, there first needs to be a trigger for the secretion of chemokines, the upregulation of adhesion factors, and the creation of an inflammatory environment in the blood vessels of the brain. This trigger could be an initial infection of the systemic flavivirus at the BBB or even a co-infection of the BBB with a different pathogen.

### 2.4. Mechanisms of Bypassing the BBB Entirely

Given the proximity of the BBB to the CNS, it is no wonder so many groups have spent time studying its relevance for neuroinvasion. However, when considering the pathway of infection, there is another important target to study. Most flaviviruses enter the human body via a mosquito or tick bite through the skin, which is studded with free nerve endings that are in direct contact with the CNS. These axonal nerve endings are believed to be used by flaviviruses to gain access to the CNS (Figure 4). Several lines of evidence confirmed this axonal transport playing a role in flavivirus infection, for example in WNV infection, which was reported to spread through motor nerve cells [88]. In vitro studies showed that WNV could spread through intact axons to target cells and other neurons, whereas if the axon was damaged, the infection was obstructed [89]. Cell-free viral particles were observed at the medium surrounding the distant axon site and neutralization of these viral particles blocked infection of target cells, thereby substantiating the hypothesis of infection with cell-free particles at the synapse [89]. In addition, viral spread by axonal transport has been reported in a significant proportion of neurotropic viruses such as Polio and Rabies, demonstrating that this mechanism is generally conserved among neurotropic virus infection [90,91].

Axonal transport can be retrograde (towards the cell body) and anterograde (from the cell body to the synapse) [92]. WNV was found to be transported in both directions in a simulation with artificial compartmentalized neurons [89]. This ability for both retrograde, as well as anterograde, transport of WNV was confirmed in vivo by ultrastructural mapping in WNV-infected Rhesus monkeys [88]. However, retrograde transport appears to be more relevant for neuroinvasion, as it provides the virus with a direct way to travel towards the CNS from the periphery without having to cross the BBB.

Although both motor and sensory neurons are present in the sciatic nerve, where the WNV infection took place, only motor neurons were involved with CNS invasion [93]. Within the brain, the WNV spread showed a preference for areas involved in motor control [88]. In accordance with this mechanism, WNV induces severe neuromuscular pathology [94]. An alternative hypothesis is that motor neuron pathology is caused by a bystander effect of the lack of glutamate, an excitatory neurotransmitter, caused by WNV-infected astrocytes, leading to hyper-excitation and glutamate toxicity [95]. However, this hypothesis was based on experiments with Sindbis virus, a different positive-stranded RNA virus.

An indication toward the intracellular mechanism of axonal flavivirus transport was provided by electron microscopy of WNV-infected Rhesus monkey neurons. There, autophagosomes containing several viruses were found at the synapse site [88]. Virus could be autophagocytosed by the cell to be degraded at the cell stoma. However, this mechanism could also be a way for the virus to take advantage of the cell machinery to be transported to the cell body and replicate there. Indeed, JEV was reported to induce autophagy in neuronal cells to accomplish replication at the cell body [96]. Additionally, autophagic vacuoles containing TBEV virions were demonstrated by electron tomography to have a connection with microtubules [97]. Treatment with nocodazole, a microtubule-destabilizing agent, greatly inhibited the released amount of infectious TBEV in vitro and delayed viral entry in the brain in vivo of both TBEV as well as WNV [97,98]. Moreover, ZIKV and DENV were reported to use autophagy as an egress mechanism by releasing membrane-enclosed infectious particles, although this was not tested in neuronal cells [99]. These reports point towards autophagocytosis as the main mechanism of flavivirus axonal transport. In conclusion, although axonal transport is an essential mechanism in flavivirus CNS entry, all of the work summarized thus far suggests that these different mechanisms of overcoming the BBB often work in parallel to allow the flavivirus to achieve efficient neuroinvasion.

## 3. Neuropathogenesis

As reported previously, flavivirus infections can provoke mild and severe neurological disease, depending on the virus and the immune status of the host [9]. To date, different mechanisms have been investigated, and this work seems to suggest, similarly to neuroinvasion, that no one mechanism can be responsible for all of these symptoms. Currently, three main causes for neuronal death are thought to have the highest influence: virus-induced, immune-induced, or bystander effects on uninfected cells.

### 3.1. Neuronal Death after Infection

The first mechanism of neuronal cell death, and thereby neurologic sequelae and mortality, is direct killing or growth restriction of the cells by the virus during infection (Figure 5A). It has been well-documented that flavivirus infection causes cell death in in vitro neuronal cultures [100,101,102]. The pathways leading to cell death however can differ greatly. Lytic cell death, or the cell “bursting” during replication of the virus, is a commonly occurring mechanism during viral infection. Interestingly, DENV and JEV were described to encode for non-structural proteins called viroporins (NS2B) that can perforate the cell membrane and increase membrane permeability [103,104]. Although not yet tested in neuronal cells, these proteins cause cytopathic effects in other cell types and therefore might also contribute to neuropathogenesis. TBEV was also reported to cause cytopathic effects, necrosis, and apoptosis in human neural cells [105].

Apoptosis is induced by a wide range of flaviviruses as reviewed by Pan et al. [106]. The widespread modulation of apoptotic pathways by flaviviruses suggests that this is another important mechanism in flavivirus-induced neuronal cell death (Figure 5A). Indeed, apoptosis was detected after WNV infection of neuronal cells as indicated by Annexin V staining, DNA fragmentation, and the TUNEL assay [107]. Moreover, apoptosis was observed as a result of JEV infection [101]. Contrarily, another report demonstrated that apoptosis was rare after TBEV and WNV infection [108]. This inconsistency could be due to primary neuron sampling from a different area in the brain, or a different time point of examination [108]. Pan and colleagues describe modulation by flaviviruses in both the intrinsic and extrinsic pathways of apoptosis [106]. The intrinsic pathway mostly relies on intracellular signals, whereas the extrinsic pathway is triggered by death receptors on the surface [106]. For instance, WNV-infected primary cortical neurons upregulated caspase 3, a protein appearing in both the intrinsic and the extrinsic pathway [109]. Another report shows WNV induction of Bax-dependent apoptosis, a protein mostly belonging to the intrinsic pathway [102]. Additionally, a neuroblastoma cell line (N2a) was reported to overexpress TNFR-1, a receptor from the extrinsic pathway, during JEV infection, as well as JNK and p53, which are proteins belonging to the intrinsic pathway [101].

Although the in vitro detection of wide-spread apoptosis in flavivirus-infected neuronal cell cultures suggests a role for apoptosis in neuronal death, it does not definitively prove a role for apoptosis in neuropathogenesis. The utilization of caspase 3 knockout mice, however, did accomplish this as they were shown to be more resistant to lethal WNV infection, even though the viral load in the brain remained stable [109]. Another group supported the role for apoptosis in pathogenesis by knocking down TRADD, a protein downstream from the TNFR-1 receptor, leading to a decrease in JEV-induced apoptosis and higher survival rates in mice [101]. However, fewer apoptotic neurons were not the only consequence of TRADD silencing. It was reported that there was less microglial and astrocyte activation, decreased leukocyte infiltration, and lower ICAM and VCAM expression on endothelial cells [110]. Therefore, it cannot be concluded with certainty that the improvement in life expectancy of TRADD-silenced mice is a direct result of a lower apoptotic rate, or due to the complicating factor of reduced apoptosis, leading to lower immune activation in general. Nevertheless, there is a large body of evidence showing that apoptosis forms an important neuropathogenic mechanism for flaviviruses.

In addition to differentiated neurons, flaviviruses can also target neural progenitor cells (NPCs), which form robust pools of stem cells that are important for maintenance in different regions of the adult brain [111]. JEV was shown to target NPCs in the subventricular zone (SVZ) of mice brains, which did not lead to direct cell death in these cells, but a blocked cell-cycle progression from G1 to S phase, leading to an inability to divide and perform their stem-cell function (Figure 5A) [112]. A similar mechanism was shown to contribute to congenital Zika syndrome, where ZIKV was reported to target NPCs, thereby attenuating their growth [113].

### 3.2. Immune System Killing Infected Neurons

Infection of the CNS does not pass unnoticed and, while the neural immune system has an important role in the clearance of viruses, it can also contribute to pathogenesis [114]. The CNS contains several lines of defense against viral intruders; the first consists of microglia, which serve as CNS-resident macrophages, and provide protection against incoming pathogens by phagocytosing them [115]. Indeed, the depletion of microglia in WNV-infected mice leads to increased viral titers and more severe disease [116]. On the other hand, it has been suggested that microglia can also contribute to neuropathogenesis in flavivirus infections. For example, one recent report demonstrated that activation of microglia can induce loss of pre-synaptic terminals of WNV-infected neurons in the hippocampus (Figure 5B) [117]. The hippocampus is responsible for the formation of memories and therefore the loss of synaptic terminals can lead to learning deficits [118]. The mechanism of this synapse elimination is described to be facilitated by the complement system, which is also involved in synapse elimination in a healthy situation during development [119]. This group further supported this claim by showing that WNV-induced synapse elimination was abrogated in microglia- or complement-deficient mice [117]. In another report, this role of microglia in synaptic terminal loss was confirmed, and T-cells were suggested to activate the process. T-cells were found to produce a trigger for microglia activation, namely, IFNɣ [120]. This conclusion is further supported by the fact that mice deficient in IFNɣ-producing T-cells or IFNɣ-signaling incompetent microglia were protected against the loss of synapses during WNV or ZIKV infection, suggesting that this mechanism consists of an interplay of several factors. Importantly, T-cells not only have a role in assisting microglial-mediated pathology. Cytotoxic T-cells also play their own role in neuropathogenesis.

After flavivirus invasion into the CNS, cytotoxic T-cells (CD8+) can invade the neural tissue via recruitment induced by chemokines and adhesion proteins. These invading CD8+ T-cells have been shown to contribute to flavivirus neuropathogenesis as they can cause damage to infected neurons, using either fas-ligand or granzyme-related pathways (Figure 5B) [121]. For instance, after infection with MVEV, a virus related to JEV, it was found that mice deficient in fas-ligand or granule exocytosis were more resistant to encephalitis [122]. Contrarily, another group reported that CD8+ T-cells offered great protection against BBB leakage and JEV-induced morbidity [123]. This discrepancy in the role of CD8+ T-cells could be related to the place the CD8+ T-cells are operating: in the periphery, they may prevent further viral production and dissemination, whereas in the CNS, they can damage non-regenerative neurons. This was confirmed by studies reporting TBEV infection in mice, where CD8+ T-cell-deficient mice showed prolonged survival to TBEV infection [124]. When CD8+ cells were transferred into the CD8+-deficient mice, the shorter survival time returned. After the mice were sacrificed, it was found that wild-type mice had more infiltrates in the brain, especially of CD8+ T-cells [124]. Additionally, the infiltration of CD8+ T-cells in the brain after TBEV infection was confirmed in human cases, where post-mortem immunohistochemistry showed CD8+ T-cells in direct contact with TBEV-infected neurons [125]. These results demonstrate that although CD8+ T-cell cytotoxicity can help prevent further viral spread by killing infected cells, it can also cause severe damage in some areas of the body, such as the immune privileged CNS. In peripheral tissues, these eliminated cells can be regenerated. However, because of the low regenerative rates of the brain, this neuronal death leads to severe and lasting neurological symptoms [126].

### 3.3. Non-Infected Bystander Neurons

The infection of neurons and subsequent immune activation does not only cause damage to infected cells, but also uninfected cells, or so-called “bystander” cells (Figure 5C). This bystander damage can, for example, result from the highly inflammatory environment that is created by the innate and CNS-intrinsic immune system [127]. For example, JEV infection has been shown to cause microglia to produce a large number of inflammatory cytokines. Microglia contain the CLEC5A, a C-type lectin receptor that is responsible for regulation of cytokine release. Interaction of JEV with CLEC5A induced proinflammatory cytokine release by macrophages and microglia [128]. When the medium taken from JEV-infected microglia was UV-irradiated to inactivate the virus and added to neurons in vitro, there was even higher neuronal death than following direct JEV addition, showing that bystander neuron damage contributes to a large part of neuropathogenesis [128]. Strengthening this hypothesis, CLEC5A knockout mice are protected from JEV-induced lethality despite continued JEV infection in neurons [128]. In addition, a higher inflammatory cytokine level in the CNS, as well as in serum, correlated with a poor outcome in DENV and JEV patients [46,129]. However, cytokines do not only directly cause bystander neuron damage; they can also cause damage by breaking down the BBB.

As discussed previously, cytokines and chemokines upregulated after flavivirus infection can result in BBB leakage [130]. BBB leakage has large-scale consequences. On the one hand, it can play an essential role in virus neuroinvasion, where it permits peripheral immune cells to enter the brain and combat the virus. On the other hand, BBB breakdown has been linked with a myriad of diseases. In the first place, BBB leakage can cause a positive feedback loop, where cytokines and chemokines that infiltrate the CNS because of the BBB leakage attract more T-cells, neutrophils, and macrophages into the CNS and activate CNS-resident immune cells, leading to enhanced inflammation in the CNS and more BBB permeability [131]. As detailed before, this increased inflammation can then lead to bystander damage, and the cyclical process repeats. Furthermore, BBB leakage is associated with acquired epilepsy as well as Parkinson’s disease [132,133]. Although BBB leakage has not been described to have a causal relationship with these diseases, it is clear that it contributes to their pathogenesis, and could have the same effect on flavivirus-related pathologies. Therefore, BBB breakdown is not only an important contributor to neuroinvasion, but also neuropathology.

## 4. Conclusions

Flaviviruses are endemic in over 100 countries and can cause severe neurological symptoms, thereby resulting in significant morbidity and mortality [134]. Neuroinvasion is an important part of flavivirus pathology for which they employ several different mechanisms. The most commonly studied is flavivirus entry by BBB breakdown. This happens, for example, by the upregulation of TJ-degrading proteins, TLR3 stimulation, or leukocyte recruitment. The second mechanism of neuroinvasion by flaviviruses consists of BBB passing without breakdown. This can happen through endothelial infection, transcytosis, or the “Trojan horse” principle. Finally, flaviviruses have also been shown to bypass the BBB entirely through axonal transport. During this phenomenon, the virus enters the axon at the periphery, after which the virus can be transported across the neuron from the periphery all the way to the CNS.

The existence of the above-mentioned neuroinvasion mechanisms has been extensively studied and discussed. However, the interplay between these mechanisms remains uncertain. Interestingly, after WNV infection, fluctuations of the viral load in the CNS over time were reported [135]. After peripheral inoculation of WNV in the footpad of mice, WNV appeared at a low viral load two days post-infection in the spinal cord, cerebellum, and brain stem. Subsequently, the virus was cleared, only for it to return at six days post-infection with a much higher viral load [135]. This suggests that the virus enters the CNS at several different time points during infection, which substantiates the parallel existence of different CNS entry pathways during the same infection. Furthermore, the entry mechanisms of leukocyte recruitment and cytokine release require a primary trigger, indicating that the study of different entry mechanisms in isolation does not provide the full picture. Indeed, in vivo data show that BBB permeability and MMP production only occur 6 days after infection, despite viral RNA already being detected in the brain, showing that although it contributes to large-scale invasion, it might not be the first mechanism [27]. Therefore, in the future, studying the appearance of and decrease in the different entry mechanisms over time will likely better represent the complete picture.

Unfortunately, several experimental difficulties complicate the study of neuroinvasion and neuropathogenesis; for example, the age of the mice used influences T-cell maintenance [136]. Moreover, the dose and delivery method are extremely important to consider. Intracranial injection, for example, does not include BBB passage and peripheral inoculation with a low dose of virus might not result in neuroinvasion at all. In addition, host species and strain- or virus-specific virulence factors may influence the appearance of neuroinvasion in a model [137,138]. These details need to be considered carefully when studying neuroinvasion and neuropathogenesis. Additionally, neural sampling at different time points in vivo is difficult and complicates in vivo studies. Therefore, human brain organoid models might offer a useful alternative for exploratory studies about neuroinvasion as well as neuropathogenesis. Recently, human brain organoid models were explored in the context of viral pathogenesis. However, these models do not contain a BBB making them unsuitable to study neuroinvasion [139]. An iPSC-derived BBB model overcomes this disadvantage, but does not recreate the 3D structure of the brain [140,141]. Therefore, despite the development of better in vitro models, there still is a need for in vivo data.

It has been hypothesized that flaviviruses not only pass through the BBB but also through the lesser-studied blood–CSF barrier, formed between the CSF in the choroid plexus and surrounding blood vessels. Additionally, reports have shown a possible invasion route through an olfactory and a gut–brain neural circuit. However, more research is needed to confirm these routes of invasion and to determine their relative contribution to neuroinvasion [142].

After flaviviruses have entered the CNS, they provoke neuronal damage. This damage occurs through direct infection of neural cells, which can succumb to infection through cell lysis, necrosis, or apoptosis. The neuronal damage can also be caused by the immune system, for instance, microglia and cytotoxic T-cells. While these cells certainly contribute to cell death, it is important to remember that several immune components perform an indispensable protective function during flavivirus infection. When studying these infections, the fragility of the balance between providing enough inflammation and immune activation to clear the infection, but not so much as to irreparably damage the brain, becomes apparent. This calls for more research about which strategies can be employed to maintain this balance and thereby limit flavivirus- or immune-induced pathology. Additionally, it is still unclear how much each one of the neuropathogenesis mechanisms contributes to clinical outcome. This is especially difficult to study as most mechanisms can aggravate each other in vivo. Therefore, future research will have to shed light onto these unknowns.

## Figures and Tables

**Figure 1 viruses-15-00261-f001:**
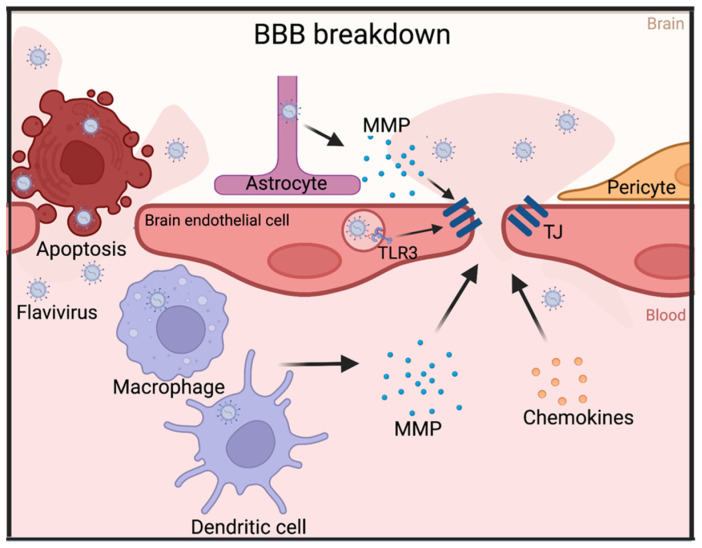
Mechanisms of BBB breakdown. Diagram demonstrating several mechanisms by which flaviviruses can induce the permeability of the blood–brain barrier. Abbreviations: MMP = matrix metallo proteases, TLR3 = Toll-like receptor 3. Made in Biorender.

**Figure 2 viruses-15-00261-f002:**
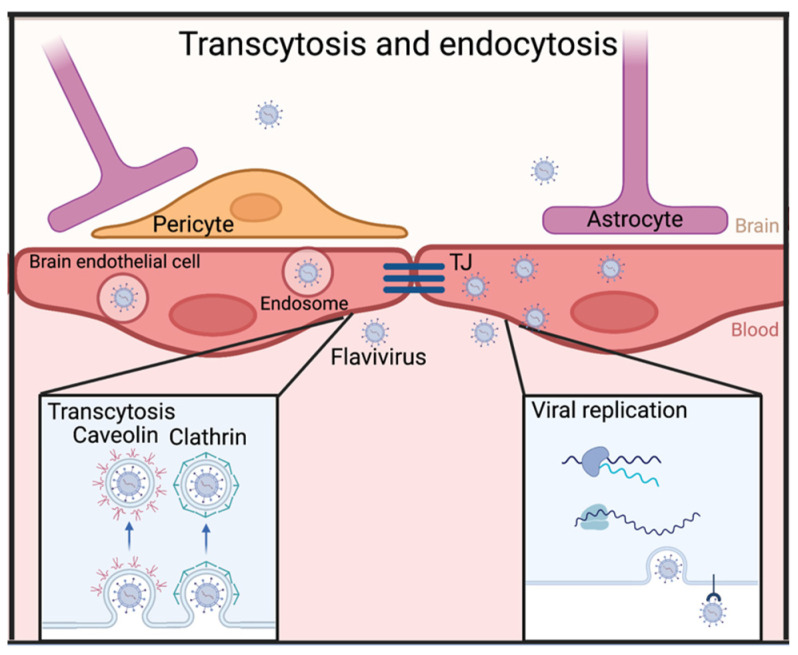
Flaviviruses bypassing an intact BBB via transcytosis and endocytosis. Made in Biorender.

**Figure 3 viruses-15-00261-f003:**
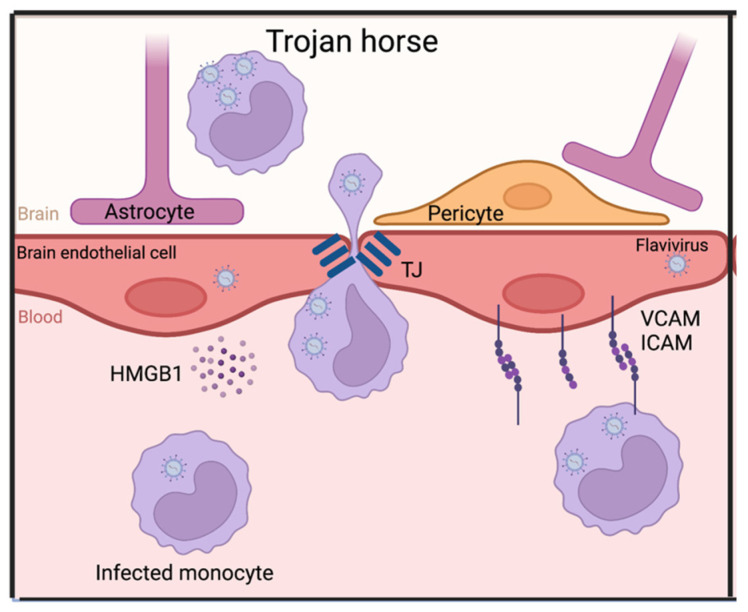
Flaviviruses bypassing an intact BBB via the “Trojan horse” principle. Abbreviations: HMGB1: High-mobility group box 1, VCAM: Vascular cell adhesion molecule, ICAM: Intercellular adhesion molecule. Made in Biorender.

**Figure 4 viruses-15-00261-f004:**
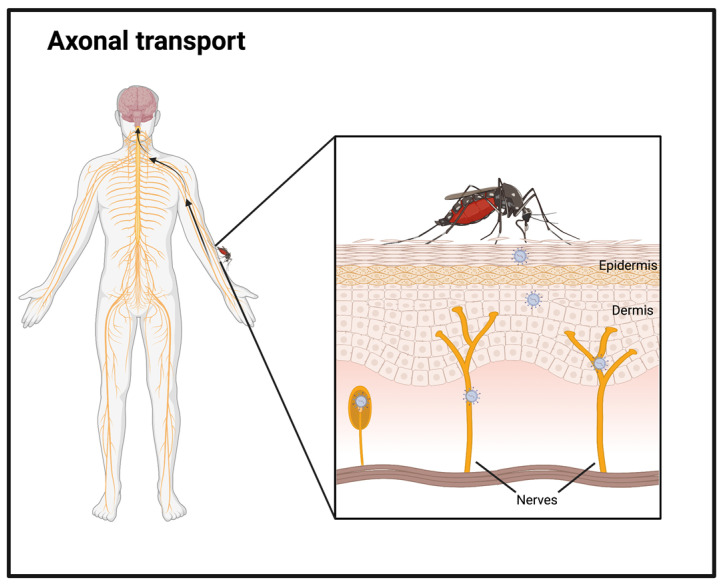
Flaviviruses bypassing the BBB via axonal transport. Made in Biorender.

**Figure 5 viruses-15-00261-f005:**
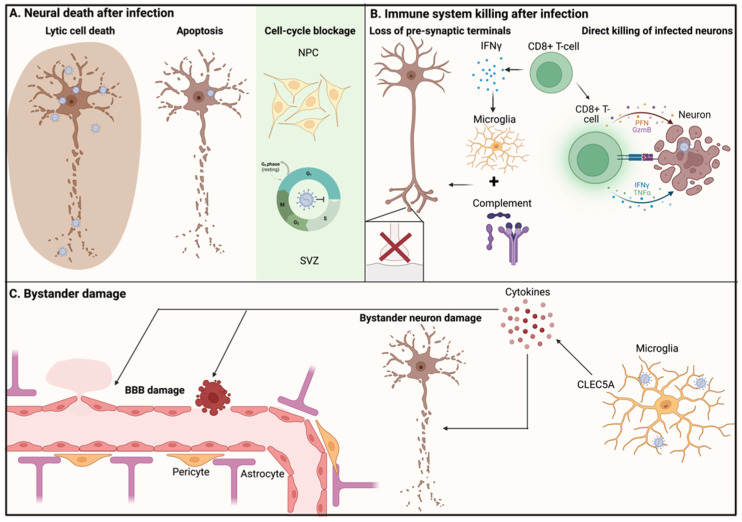
Mechanisms of neuropathogenesis by Flaviviruses. Legend: NPC = neuroprogenitor cell, SVZ = Subventricular zone, BBB damage = blood–brain-barrier damage, IFNγ = Interferon γ, CLEC5A = C-type lectin member 5A. Created in Biorender.

## Data Availability

No new data were created or analyzed in this study. Data sharing is not applicable to this article.

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
