# Peer review of "Mechanisms of Neuroinvasion and Neuropathogenesis by Pathologic Flaviviruses"

_viruses, 2023, doi:10.3390/v15020261_

Round 1

Reviewer 1 Report

Liset de Vries and Alfred T. Harding processed and analyzed a large amount of accumulated information about the interaction between flaviviruses and the central nervous system. The authors presented several hypothetical mechanisms by which various flaviviruses enter the CNS, as well as mechanisms of neuropathogenesis. Despite the large amount of available data, the mechanism of penetration of neurotropic flaviviruses into the CNS is not completely understood. Nevertheless, this work is a good foundation for researchers who work in this field.

Links to more recent work were missing in some places. However, this does not diminish the importance of this study. Also, a very similar review on this topic has recently been released, perhaps you need to add some information from it and refer to it: Marshall EM, Koopmans MPG, Rockx B. A Journey to the Central Nervous System: Routes of Flaviviral Neuroinvasion in Human Disease. Viruses. 2022 Sep 21;14(10):2096. doi: 10.3390/v14102096. PMID: 36298652; PMCID: PMC9611789.

Minor comments:

1.       Line 25-26. It should be added that these are mosquito-borne viruses.

2.       Line 53: repeat of line 31.

3.       Line 183-184: Perhaps it would be better to move this sentence to line 182.

4.       Line 241 -243 Clarify that it is for JEV

5.       Line 289 Decipher the abbreviation DHA

6.       Line 294-303 It also may be due to the peculiarities of the interaction of a certain virus strain with the immune system.

7.       Line 310 What about TBEV?

8.       Line 322 Decipher the abbreviation BFA

9.       Line 446 Missing space between caspase 3

Author Response

We would like to thank the reviewer for their insightful comments and have addressed all of their points as described below. 

  1. Corrected in text.
  2. Line 31 was deleted to eliminate the repeat.
  3. We did not decide to move this line of text as it works well with the following sentences, but we do understand the text as written was clunky. We have added additional text surrounding this area to make the transitions smoother. 
  4. Corrected in text.
  5. Added in text.
  6. We have added additional text to clarify this potential. 
  7. We have added an additional citation to include TBEV.
  8. Added in text. 
  9. Corrected in text. 

Reviewer 2 Report

The review article „Mechanisms of neuroinvasion and neuropathogenesis by pathologic flaviviruses” describes the different mechanisms how flaviviruses can enter the brain, comprising disruption of the blood-brain barrier (BBB), mechanisms that are not disrupting the BBB and bypassing the BBB by axonal transport. In the last part of the review, the authors summarize the knowledge about neuropathogenesis. This part explains the relevance of apoptosis of infected neuronal cells, the role of the immune system in killing of infected cells as well as the role of non-infected bystander neurons.

The presented review is well written and interesting to read. I recommend the manuscript for publication with only very few minor changes:

1. line 31 is repeated in line 53. Please delete one sentence.

2. Please check whether all abbreviations are introduced and used correctly.
E.g. the abbreviation “tight junctions (TJ)” is introduced twice (line 87 and line 219). After introduction of the abbreviation, the abbreviation should be used consequently: Please replace “tight junction” by “TJ” in the subsequent text.

3. Figures 1-5: Please explain all structures shown in your figures. This could be done by labelling in the figures and more detailed description in the legend.

E.g. What is the large inverted “T”-shaped structure in the center of Figure 1? Please indicate endothel cells, immune cells, flavivirus, endosome, tight junctions, ...etc.

Author Response

We would like to thank the reviewer for their insightful comments, we have addressed all of their points as detailed below. 

  1. Line 31 was deleted below to prevent the repeat.
  2. We have ensured that all abbreviations are detailed and after the abbreviation was presented it was used. 
  3. We have uploaded new figures with much detailed labelling that should make figure interpretation easier.